# Acidic graphene organocatalyst for the superior transformation of wastes into high-added-value chemicals

Aby Cheruvathoor Poulose [1], Miroslav Medveď [1,2], Vasudeva Rao Bakuru[3], Akashdeep Sharma[4], Deepika Singh[5], Suresh Babu Kalidindi [6], Hugo Bares [1,9], Michal Otyepka [1,7], Kolleboyina Jayaramulu[4] ✉, Aristides Bakandritsos [1,8] ✉ & Radek Zbořil [1,8] ✉

Our dependence on finite fossil fuels and the insecure energy supply chains have stimulated intensive research for sustainable technologies. Upcycling glycerol, produced from biomass fermentation and as a biodiesel formation byproduct, can substantially contribute in circular carbon economy. Here, we report glycerol's solvent-free and room-temperature conversion to high-added-value chemicals via a reusable graphene catalyst (G-ASA), functionalized with a natural amino acid (taurine). Theoretical studies unveil that the superior performance of the catalyst (surpassing even homogeneous, industrial catalysts) is associated with the dual role of the covalently linked taurine, boosting the catalyst's acidity and affinity for the reactants. Unlike previous catalysts, G-ASA exhibits excellent activity (7508 mmol g$^{-1}$ h$^{-1}$) and selectivity (99.9%) for glycerol conversion to solketal, an additive for improving fuels' quality and a precursor of commodity and fine chemicals. Notably, the catalyst is also particularly active in converting oils to biodiesel, demonstrating its general applicability.

Our dependence on finite fossil fuel reserves for energy, and chemicals, the high associated environmental impact, and the repeating crises in oil prices, exacerbated by the recent disruptions in global supply chains[1,2], have spurred intensive research for alternative, eco-friendly, and sustainable energy carriers and chemicals[3–5]. Concerted efforts are globally focused on transforming renewable feedstocks (carbon dioxide[6], methane[7], ethanol[8], glycerol[6], and others[9–11]) to added-value and technologically important compounds. Large amounts of glycerol are generated by producing biodiesel[12,13] as a 10 vol.% side-product during the transesterification of triglycerides from animal, vegetable, and algae oils[1,14]. It can also be made from biomass fermentation and as a byproduct of propylene synthesis or soap production[15,16]. Such activities create a sustainable supply of glycerol, rendering it an attractive renewable carbon source if effective methods for its upgrade are identified. As a result, profound attention is concentrated on glycerol's valorization towards high-value chemicals via

[1]Regional Centre of Advanced Technologies and Materials, Czech Advanced Technology and Research Institute (CATRIN), Palacký University in Olomouc, Šlechtitelů 27, 783 71 Olomouc, Czech Republic. [2]Department of Chemistry, Faculty of Natural Sciences, Matej Bel University, Tajovského 40, 974 01 Banská Bystrica, Slovak Republic. [3]Materials Science and Catalysis Division, Poornaprajna Institute of Scientific Research, Bangalore Rural, India. [4]Hybrid Porous Materials Laboratory, Department of Chemistry, Indian Institute of Technology Jammu, Nagrota Bypass Road, Jammu, Jammu and Kashmir 181221, India. [5]Quality Management & Instrumentation Division, CSIR-Indian Institute of Integrative Medicine, Canal Road, Jammu, Jammu and Kashmir 180001, India. [6]Central Tribal University of Andhra Pradesh, AU PG Centre, Kondakarakam Village, Vizianagaram, India. [7]IT4Innovations, VŠB - Technical University of Ostrava, 17. listopadu 2172/15, Ostrava-Poruba 70800, Czech Republic. [8]Nanotechnology Centre, Centre of Energy and Environmental Technologies, VŠB–Technical University of Ostrava, 17. listopadu 2172/15, Poruba 708 00 Ostrava, Czech Republic. [9]Present address: Lepty, 14 avenue Pey-Berland, 33600 Pessac, France. ✉e-mail: jayaramulu.kolleboyina@iitjammu.ac.in; a.bakandritsos@upol.cz; radek.zboril@upol.cz

dehydration, hydrogenolysis, esterification, (electro)oxidation, and acetalization routes[6,17-28].

Glycerol derivatives, such as ethers, esters, diols, and acetals, are important synthons for various industrial processes related to fuels, plastics, and fine chemicals. Glycerol's acetalization to solketal (a branched oxygen-containing compound; (2,2-dimethyl-1,3-diox-olan-4-yl)methanol) has gained considerable interest because of its broad application in cosmetics, pharmaceutics, food additives, polymers, tobacco, and petrochemicals[29]. Importantly, it is primarily applied as an additive in gasoline, diesel, and biodiesel because solketal reduces gum formation improving octane rating and anti-knocking properties[30,31]. Solketal, when mixed with standard diesel fuel, decreases uncontrolled emissions of carbon monoxide, hydrocarbons, particles, and toxic aldehydes[32]. In addition, it improves biodiesel's viscosity and cold flow properties and helps achieve the critical flash point and oxidation stability for long-term storage[33].

Solketal production is typically performed by homogeneous, non-recyclable acid catalysts, such as sulfuric acid, hydrochloric acid, or p-toluenesulfonic acids, via the acetalization of glycerol with acetone[34,35]. To harness the benefits of heterogeneous catalysis, Brønsted and Lewis acid catalysts (zeolites, metal-substituted mesostructured silica, zirconia, mixed metal oxides, metal phosphates, and sulfonic acid-functionalized carbons, resins, and polymers) are intensively studied[36] (Supplementary Table 1). For example, $SO_4^{2-}/SnO_2$ solid-state acid showed 95% glycerol conversion with 96% solketal selectivity at room temperature (Supplementary Table 1, entry c);[37] however, its specific productivity[38] (mass-normalized rate of product formation) was 52 mmol g$^{-1}$ h$^{-1}$, which is 8-fold lower than that of sulfuric acid (Supplementary Table 1, entry θ). Sulfonated mesostructured silicas achieved specific productivity quite close to sulfuric acid, albeit at 70 °C (Supplementary Table 1, entry δ)[31]. Mesoporous substituted silicates (Hf-TUD)[39] at 80 °C showed 52% solketal yield with specific productivity of 39 mmol g$^{-1}$ h$^{-1}$ (Supplementary Table 1, entry a), and acidic carbon-based catalysts[40] delivered good yields and specific productivities, however, still, substantially lower than that of sulfuric acid (Supplementary Table 1, entry ζ). Although such catalysts are fascinating in terms of their reusability, significant challenges remain because their production rates are substantially lower than those obtained using the industrial benchmark catalyst of $H_2SO_4$[34]. In contrast, poor thermal stability, limited recyclability, use of hazardous solvents, and need for high temperatures pose further limitations (Supplementary Table 1).

Recently, graphene-based materials generated a new thrust in the field of acid catalysts because of their chemical inertness, tunable electrical and thermal conductivities, and low density. Graphene's sulfonation creates Brønsted acid sites turning it into a promising solid acid catalyst for the hydrolysis of cellulose or carbohydrates toward industrially essential chemicals. Sulfonic acid-modified graphene oxide was used to transform hexoses into levulinic acid[41], for synthesizing benzimidazole[42], or as an ion-exchange material for electrochemiluminescence analysis[43]. Sulfonic acid-functionalized reduced graphene oxide was studied for acetic acid's esterification with butanol and benzaldehyde's acetalization with ethylene glycol[44]. However, to date, effective solid-state carbon- or metal-based heterogeneous catalysts for the technologically important glycerol valorization with substantially improved performance compared to the $H_2SO_4$ benchmark catalyst remain elusive. Thus, considerable efforts are required to develop efficient and stable heterogeneous Brønsted-acid catalysts with a strong potential for replacing sulfuric acid, which remains one of the most efficient and cost-effective catalysts for this reaction (Supplementary Table 1).

To tackle this challenge, we developed a graphene catalyst functionalized with a natural and abundant amino acid (taurine, 2-aminoethanesulfonic acid) for the solvent-free and room-temperature chemical conversion of glycerol to solketal with an activity several-fold higher than the sulfuric acid or other industrially used homogeneous catalysts, such as p-toluenesulfonic acid. Taurine's conjugation on graphene is achieved via the nucleophilic attack of its amino group on the electrophilic centers of fluorographene (FG), affording the covalent aminosulfonic acid-derivatized graphene (G-ASA) via a simultaneous sulfonation and defluorination process. The G-ASA catalyst, obtained without any size-selection process, directly from the reaction of the sonicated bulk graphite fluoride (GrF), afforded high glycerol conversion of 99.9 % and solketal selectivity of 96.3 %, giving a five-fold higher specific productivity (2168 mmol g$^{-1}$ h$^{-1}$) than $H_2SO_4$ (400 mmol g$^{-1}$ h$^{-1}$). The activity of G-ASA reached an unprecedented turnover frequency of 1735 h$^{-1}$ (corresponding to specific productivity of 7508 mmol g$^{-1}$h$^{-1}$) at 0.1 mass% catalyst loading (Supplementary Table 1) and displayed excellent reusability. Increasing the potency of Brønsted-acid organocatalysts has been intensively pursued by various molecular architectures and chemical group synergies[45-49]. In the case of G-ASA, density functional theory (DFT) calculations unveiled the key behind the activity of the G-ASA catalyst, lying in the dual role of taurine's protonated amino group in synergy with the sulfonic acid group. These positively charged ammonium groups in the protonated G-ASA catalyst boost the binding of the reagents and, at the same time, substantially increase the acidity of the sulfonic group, facilitating the proton exchange reaction steps very effectively. We also show that the properties of the G-ASA catalyst, endowed by its sophisticated yet one-step and cost-effective design, have broad applicability and high potency in other acid-catalyzed reactions, such as for the production of biofuels from the esterification of fatty acids.

## Results and discussion
### Physicochemical characterization
G-ASA was synthesized from GrF after its exfoliation to FG by sonication in DMF and then reacted with taurine at 130 °C under basic conditions (Fig. 1a). Fourier-transform infrared (FT-IR) spectroscopy of the starting GrF (Fig. 1b) showed bands of the CF and CF$_2$ bonds at 1200 cm$^{-1}$ and 1305 cm$^{-1}$, respectively[50], while, after the reaction, the SO$_3$H group in G-ASA gave rise to the characteristic bands at 1190 cm$^{-1}$ and 1035 cm$^{-1}$ corresponding to symmetric O = S = O and SO$_3^-$ stretching, respectively[51,52]. The broad nature of the 1190 cm$^{-1}$ band is attributed to the contribution of the evolved sp$^2$ carbon network, which also gives rise to the 1569 cm$^{-1}$ band[53]. Any contribution from CF groups around 1200 cm$^{-1}$ in G-ASA is excluded since X-ray photoelectron spectroscopy (XPS) confirmed that almost all F atoms had been eliminated (Supplementary Fig. 1). A broad feature in the region between 3600 cm$^{-1}$ and 2900 cm$^{-1}$ arises from the presence of N-H and C-H groups and H-O-H molecules.

The Raman spectrum of G-ASA (Fig. 1c) showed two characteristic graphene vibrations, the G-band at around 1580 cm$^{-1}$ (vibration of $E_{2g}$ symmetry in graphene) and the D-band at 1350 cm$^{-1}$ due to aromatic ring vibrations adjacent to sp$^3$ carbon centers bonded with taurine and other defects. Moreover, the 2D band at around 2670 cm$^{-1}$ from the overtone of the D-band is attributed to the double resonance transition in few-layered graphene, which is only Raman active in the presence of defects, e.g., surface functionalization in this case. The broad character of the D-band and the high $I_D/I_G$ ratio of 1.21 suggest the high functionalization degree in G-ASA[54].

Thermogravimetric analysis (TGA) of G-ASA revealed the presence of surface organic species. The mass loss above 200 °C (with maximum at 370 °C) is attributed to the loss of sulfur-containing covalently-bonded moieties due to taurine's sulfonic acid functional groups, as verified by the emission of SO and SO$_2$ gasses (Fig. 1d) and further confirmed by X-ray photoelectron spectroscopy (XPS) analysis by the elimination of sulfur from G-ASA after its thermal treatment at

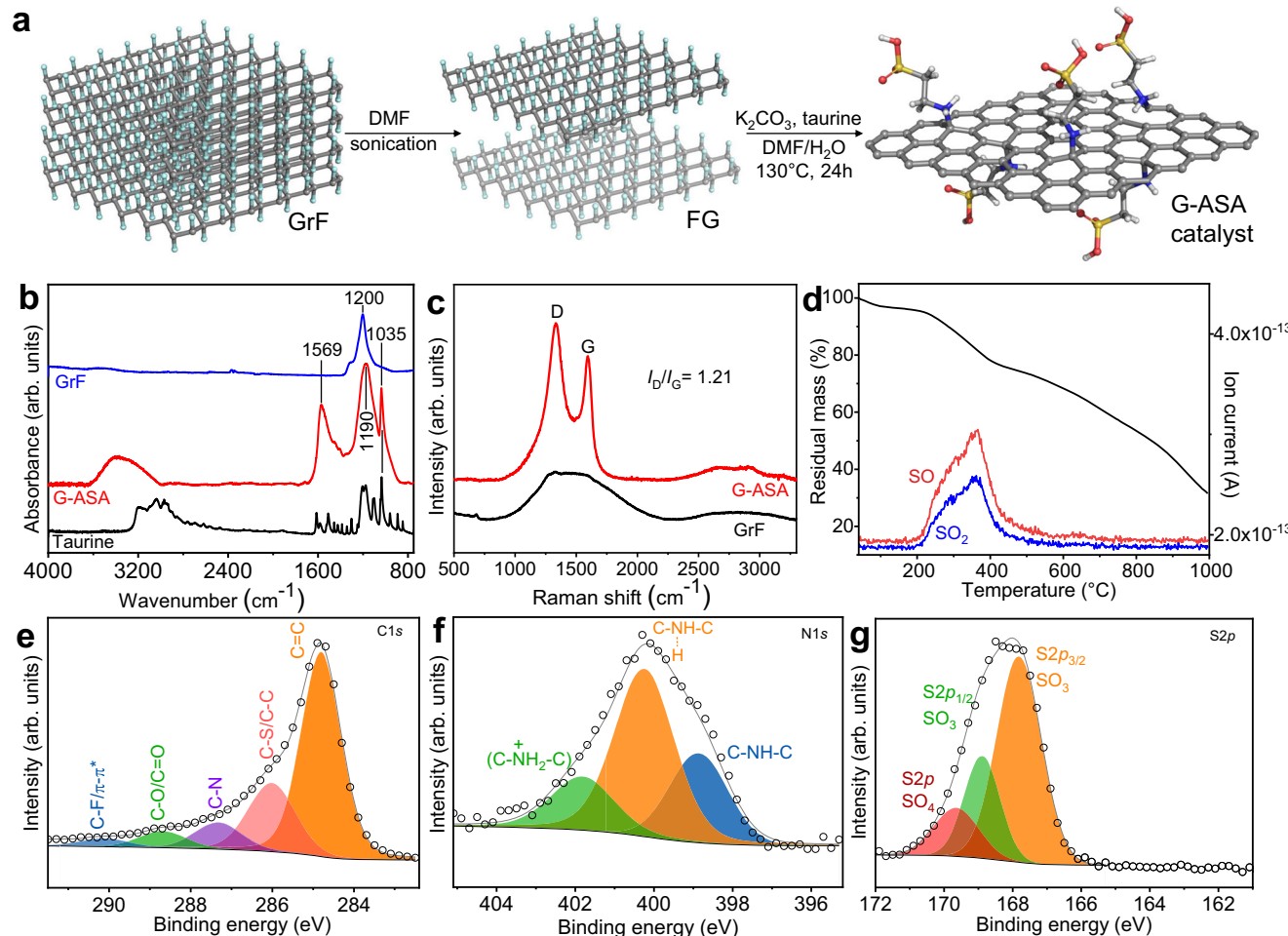

**Fig. 1 | Synthesis and chemical identity of the G-ASA catalyst. a** The synthetic route toward G-ASA from fluorographene (FG) and taurine (2-aminosulfonic acid). **b** FT-IR spectra of GrF, taurine and the G-ASA product. **c** Raman spectra of GrF and G-ASA, (**d**) TGA with evolved gas analysis for the G-ASA, and deconvoluted XPS spectra of the G-ASA catalyst for the regions of (**e**) C 1s, (**f**) N 1s, and (**g**) S 2p.

500 °C (Supplementary Fig. 1). Based on the mass loss between 200 and 550 °C, the mass of taurine groups in the sample was 24.4 mass % (and 71.2 % carbon after 500 °C), corresponding to 2.1 mmol g$^{-1}$ of SO$_3$H and 1 sulfonic acid (or 1 taurine) unit per 28.7 carbon atoms of the graphitic skeleton, indicating a functionalization degree of 3.5 %. The zeta potential of G-ASA was −35.5 mV at pH 3.4 and −36.5 mV at pH 8, showing the strongly negatively charged surface even at low pH values due to the low acidity, as later discussed. The XPS-based (Supplementary Fig. 1a) content in SO$_3$H acidic sites was 2 mmol g$^{-1}$, which is in agreement with the TGA results and with the total acid density of 3.9 mmol g$^{-1}$, obtained from the acid-base titration, since the protonated G-ASA catalyst bears an equal amount protonated amino-groups acting as extra acidic sites. The XPS spectrum of the C 1s region of the fresh catalyst (Fig. 1e) is also in agreement with the G-ASA structure, while the slight excess of N and O (Supplementary Fig.1a) comes from the reaction in DMF (as typically observed in FG chemistry[55]). The N 1s core level XPS spectrum (Fig. 1f) showed three components at binding energies of 399, 400.1, and 401.6 eV, assigned to the secondary non-protonated amine (C-NH-C), to the related hydrogen bonding configurations[56], and the protonated[55] secondary amine groups, respectively. The N 1s XPS core level spectrum of pure taurine (Supplementary Fig. 2) shows a substantial shift for all three N-components at higher eVs in comparison to G-ASA, indicative of the lower electron density of the primary nitrogen in taurine in comparison to the secondary nitrogen in G-ASA, thus confirming the covalent conjugation of taurine to the graphene support.

Transmission electron microscopy (TEM) analysis revealed few-layered, transparent flakes with a lateral size of ca. 1 μm (Fig. 2a). Higher-resolution images confirmed a disordered structure due to the high functionalization degree and the presence of aliphatic (taurine) groups at the surface of graphene (Fig. 2b, c). Energy-dispersive X-ray analysis (EDS, inset Fig. 2d) and elemental mapping with high-angle annular dark-field scanning transmission electron microscopy (HAADF-STEM, Fig. 2d–h) confirmed the presence and homogeneous distribution of the C, N, O, and S elements throughout the graphene layers.

### Catalytic activity and theoretical studies

The catalytic activity of G-ASA was tested for the conversion of glycerol to solketal at room temperature (Fig. 3a). Analysis of the reaction products showed 96.5 % glycerol conversion and 96.8 % selectivity for solketal, corresponding to specific productivity of 2094 mmol g$^{-1}$ h$^{-1}$ (Fig. 3b and Supplementary Table 1, entry #3). To probe the activity of the catalyst, we performed the reaction with low catalyst loading (0.1 mass%), affording specific productivity of 7508 mmol g$^{-1}$ h$^{-1}$ and a TOF value of 1735 h$^{-1}$ (Fig. 3b and Supplementary Table 1, entry #1). Glycerol conversion even reached 99.9 % by decreasing the glycerol: acetone mole ratio to 1:2 (Fig. 3b).

The recyclability and stability of the G-ASA catalyst were investigated by recovering and reusing the catalyst for glycerol acetalization three times without loss of any activity and selectivity (Fig. 3c). XPS analysis of the used catalyst after three reactions (Supplementary Fig. 3) confirmed the preservation of its structural features due to the

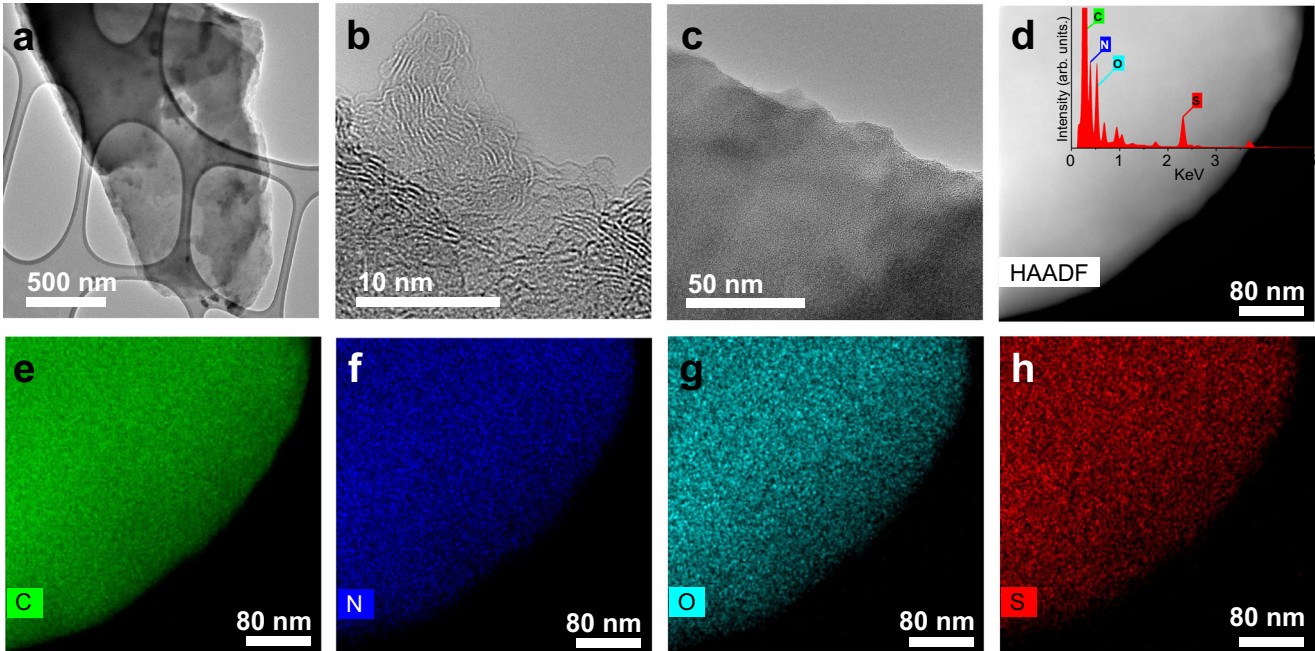

**Fig. 2 | Structural identity of G-ASA catalyst. a–c** High-resolution transmission electron micrographs of few-layered flakes of the G-ASA catalyst. **d** High-angle annular dark-field scanning transmission electron micrographs of the flake and the corresponding energy-dispersive X-ray analysis spectrum (inset), along with X-ray chemical mapping for (**e**) carbon, (**f**) nitrogen, (**g**) oxygen, and (**h**) sulfur.

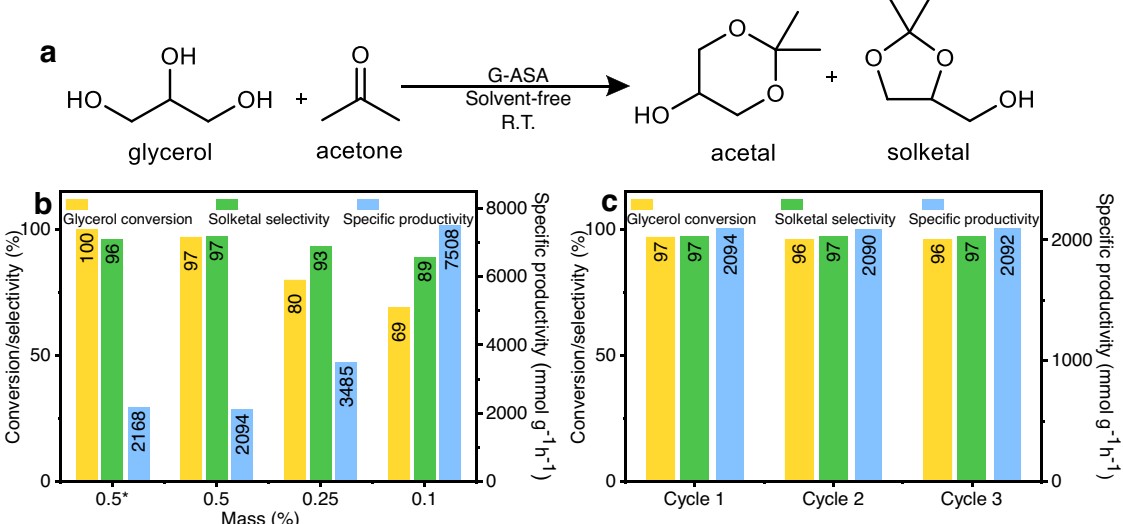

**Fig. 3 | Catalytic reaction study. a** Acetylation reaction of glycerol with acetone. **b** Solketal synthesis from glycerol by using the G-ASA catalyst (taurine-functionalized graphene) with different catalyst mass loadings (mass %) and (**c**) recyclability. Reaction conditions: glycerol = 1.0 g, acetone = 2.52 g, moles$_{glycerol}$:moles$_{acetone}$ = 1:4 (*1:2), catalyst (G-ASA) = 0.5 mass% (with respect to glycerol substrate), reaction carried at R.T. for 1 h.

covalent nature of taurine immobilization on graphene. To further check the stability and heterogeneity of the catalyst, we performed a leaching test, whereby the catalyst was separated from the reaction mixture after 5 min from starting the reaction, after which point no further glycerol conversion was observed by GC, confirming that there is no leaching of any catalytically active species from the catalyst's surface in the reaction mixture.

We executed a time-resolved investigation to evaluate the evolution of product selectivity and reaction rate over time (Supplementary Fig. 4). Results indicate a progression in glycerol conversion from 64.7% at the initial 30 min to 96.5% at 60 min, followed by a decline to 92.5% and solketal selectivity of 92.1% after 120 min of reaction.

Similarly, the reaction rate (specific productivity) also exhibited a decline. The observed decrease in conversion is attributed to the hydrolysis of products by water, which is formed during the reaction, while the drop in reaction rate is due to the depletion of reactants.

To gain insight into the high catalytic activity of G-ASA and understand the specific role of a taurine moiety in the process, we studied the binding of the reagents to the G-ASA catalyst. We also analyzed the reaction intermediates (Fig. 4) by DFT calculations at the ωB97XD/6-31 + G(d)/SMD(solvent=acetone) level[57–59], using the finite-size model of G-ASA (Supplementary Fig. 5 and 6A). In the acidic environment, the G-ASA catalyst efficiently binds the reactant molecules via multiple hydrogen bonds with binding energies of ca.

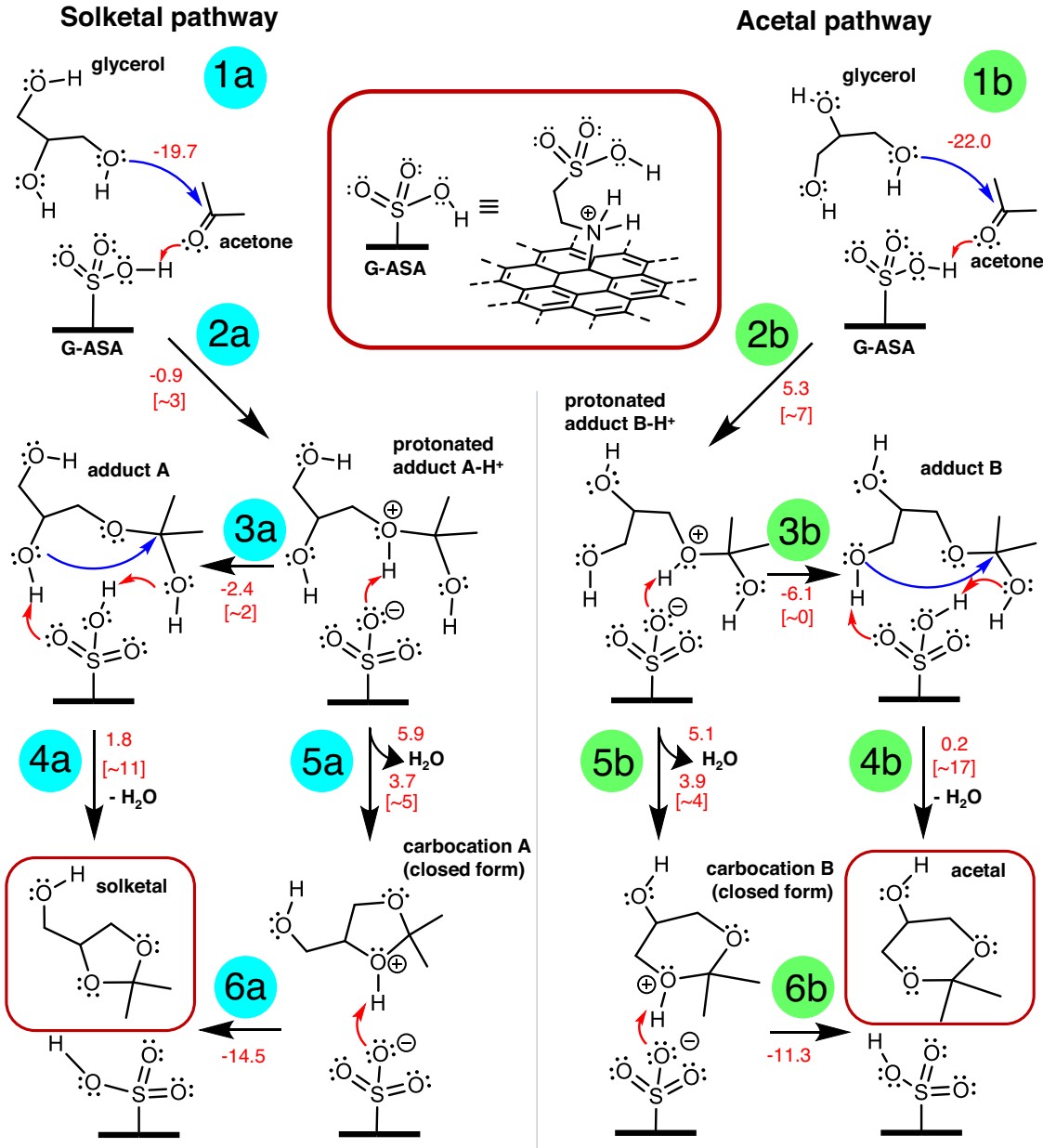

**Fig. 4 | Mechanism of the catalytic reaction.** Plausible mechanistic solketal (left) and acetal (right) pathways of the acetalization of glycerol with acetone on the G-ASA catalyst. Electronic reaction energies and activation barriers (in parentheses) obtained at the ωB97XD/6-31 + G(d)/SMD level of theory are in kcal/mol. Blue and red arrows indicate nucleophilic attacks and proton transfers, respectively.

20–22 kcal/mol (step 1). The formation of a glycerol-acetone adduct is facilitated mainly by the presence of protonated sulfonic groups on the catalyst, enabling effective proton exchange between the reagents and the substrate. While the activation barrier of this fusion reaction is about 23 kcal/mol without the catalyst (Supplementary Fig. 7), it dramatically drops to ca. 3 and 7 kcal/mol in the presence of G-ASA (steps 2a and 2b, respectively). In particular, the catalyzed reaction starts with the protonation of the carbonyl oxygen of acetone and subsequent nucleophilic attack of the hydroxyl group of glycerol on the carbon attached to the protonated oxygen (Supplementary Fig. 8–11). A thermodynamically favorable and kinetically feasible back proton transfer from the protonated adduct to the catalyst closes the first phase of the reaction (step 3). Although energetics favor the solketal path, the differences between the two pathways are relatively small, and thus the adduct formation is probably not a step determining the selectivity of the catalyst. It is worth noting that the protonated

adducts represent local minima on the potential energy surface and thus could eventually lead to the products (Fig. 4, steps 5 and 6, Supplementary Fig. 12, Supplementary Table 2). However, the practically barrierless proton transfers (step 3) suggest that the formation of products starts from non-protonated species. The release of a water molecule in the cyclization phase (step 4), beginning with a nucleophilic attack of a hydroxyl group from the glycerol moiety on the acetonic carbon, is also facilitated by the effective proton exchange with G-ASA (Supplementary Fig. 13 and 14). Whereas a five-membered ring (solketal) is formed via a nucleophilic attack by the adjacent (secondary) -OH group with the activation barrier of -11 kcal/mol (step 4a), a six-membered ring (acetal) is produced through the attack of a terminal (primary) -OH group with the barrier of -17 kcal/mol (step 4b). The higher barrier for the acetal pathway and the higher thermodynamic stability of solketal product compared to acetal (by 2.9 kcal/mol at the applied level)[60] can thus be the reasons for the higher

selectivity of the catalyst towards the solketal formation. Indeed, as we can see in the comparative Supplementary Table 1, selectivity is rarely an issue, but mainly the activity of the catalyst.

To tackle this point, we performed DFT calculations for the alkyl sulfonic acid-based catalyst (Supplementary Fig. 6B), where the amino group of taurine was replaced with a methylene group, which unveiled the dual role of the amino group of the G-ASA catalyst. First, the positive charge associated with the ammonium groups significantly increases the binding affinity of the reaction site towards the reagents, thus enabling their efficient collisions and suitable arrangement for the reaction (Supplementary Fig. 9B and 11B). Indicatively, substituting the ammonium group with the methylene group leads to an almost twofold decrease (from 20–22 to 13 kcal/mol) in the total binding energies of glycerol and acetone. Second, the presence of the ammonium group leads to a profound increase in the acidity of the catalyst (with $pK_a$ lower by ~8 units in comparison to the methylene-substituted analog, Supplementary Table 3) due to the electron-withdrawing effect of the ammonium group, eventually weakening of the electron density of the oxygens in the sulfonate moieties. The high acidity greatly facilitates the proton exchange between the substrate and reagents/intermediates. As a result, the reaction energy profiles for the catalyst with and without the amino group in the structure clearly show consistently lower energies for all reaction steps for the G-ASA catalyst (with taurine) compared to the alkyl sulfonic acid catalyst analog (Supplementary Fig. 15).

### Current technology placement and broader impact

Due to the synergy between the acidic sulfonic group and the dual role of the amino group for improving the binding of the reactants and profoundly increasing the acidity of the catalyst, the G-ASA catalyst outperforms in terms of activity previously reported Brønsted or Lewis acid catalysts (Supplementary Table 1 and Fig. 5), although the reaction was performed at room temperature. For a clear comparison, we considered TOF (where active sites were possible to determine) and specific productivity values, which rely on the total mass of the catalyst used in the reaction (Supplementary Table 1). According to this analysis, the present G-ASA catalyst shows excellent production rates and TOF values, with yields and selectivities above 90 % (Supplementary Table 1), keeping full reusability. To view these results in a broader context, sulfonic acid-functionalized mesostructured silicas[31] have shown excellent catalytic activity in the acetalization of glycerol with specific productivity of 347 mmol g⁻¹h⁻¹ (Supplementary Table 1, entry δ). However, higher reaction temperature (70 °C) and excess catalyst loading (5 mass %) were required for 80% glycerol conversion. In another example, Mo-doped SnO₂-based solid acids[24] showed intermediate catalytic activity with specific productivity of 103 mmol g⁻¹h⁻¹ (Supplementary Table 1, entry b) due to low glycerol conversion (71 %) and high catalyst loading (5 mass %). Homogeneous and industrially used catalysts such as sulfuric acid (specific productivity 400 mmol g⁻¹h⁻¹) and p-toluene sulfonic acid[34] (specific productivity 251 mmol g⁻¹h⁻¹) show high activities for glycerol acetylation (still significantly lower than G-ASA), but they are not recyclable and produce toxic byproducts, posing additional environmental concerns. Sulfonated carbon[25], sulfonic acids functionalized activated carbon[61], and acidic carbon[40] are indicative examples of sustainable and eco-friendly catalysts for glycerol acetalization, but their specific productivities are low, despite having high acid densities. The present G-ASA catalyst could convert glycerol very efficiently at room temperature by using very low catalyst loading (0.25 mass %). Thus, the specific productivity was 3485 mmol g⁻¹h⁻¹, reaching 7508 mmol g⁻¹h⁻¹ when challenged at the limits of its activity, at catalyst loadings of 0.1 mass %, offering not only a possible recyclable alternative for H₂SO₄ but also a catalyst with much higher performance.

To demonstrate the broader applicability and scope of G-ASA, we studied the esterification of fatty acids, one of the industrially

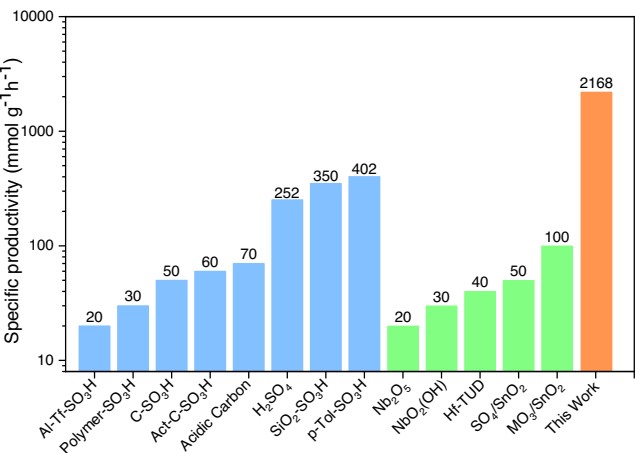

**Fig. 5 | The catalyst performance compared to state of the art.** Specific productivity of G-ASA catalyst and previously reported glycerol acetylating heterogeneous Brønsted acid (blue) and Lewis acid (green) catalyst. (C-SO₃H[25], Act-C-SO₃H[61], Al-Tf-SO₃H[66], SiO₂/SO₃H[31], Polymer-SO₃H[67], Acidic Carbon[40], H₂SO₄[34], p-Tol-SO₃H[34], Hf-TUD[39], MO₃/SnO₂[24], SO₄/SnO₂[37], NbO₂(OH)[68], Nb₂O₅[69]).

important biodiesel production reactions predominantly performed with acidic catalysts[62]. Esterification of palmitic acid and stearic acid with methanol showed complete conversion and 100% selectivity (based on NMR, Supplementary Fig. 16) after a 4 h reaction at 60 °C. To exclude any role of metal impurities in G-ASA, we also performed control experiments with graphenes (such as cyanographene and FG) under similar conditions, with conversions lower than 1%, although the metal contents were higher to those in the G-ASA catalyst (Supplementary Table 4). The performance of G-ASA substantially outperforms most previously reported catalysts, although higher temperatures or longer reaction times were often used (Supplementary Table 5). For example, Toda et al. reported 100 % yield in the esterification of oleic acid and stearic acid at 4 h and 80 °C using sulfonated amorphous carbon derived from sugars[62]. More recently, the esterification of oleic acid based on a sulfonated magnetic solid acid catalyst showed a 99.5% yield for a 4-h reaction at 90 °C[63]. Moreover, the G-ASA catalyst showed excellent stability in this reaction, even after five consecutive catalytic cycles, as evidenced by the XPS analysis of the five times used and recovered catalyst (Supplementary Fig. 17).

We report a particularly effective catalyst for glycerol's selective and remarkably swift transformation and valorization to solketal using a previously unexplored graphene derivative. Covalently bound taurine molecules on graphene (G-ASA) yield an aminosulfonated graphene with a very high (40 mass%) content in taurine, which could catalyze the acetylation of glycerol more effectively even than homogeneous and very effective industrial catalysts. The presence of the amino group dramatically boosted the acidity of taurine's SO₃H active site, and significantly enhanced the binding affinity of the reactants, promoting proton exchange and intermediates formation, leading to excellent catalytic activity. The high catalytic activity, superior acidity, recyclability, and simple, cost-effective synthesis of the G-ASA predispose it as a suitable and general catalyst for other critical acid-catalyzed chemicals, including biodiesel production through esterification. These findings revealed a previously ignored strategy for boosting the heterogeneous and solvent-free acetalization and esterification towards high-value chemicals and (bio)fuels.

## Methods
### Materials and reagents
Graphite fluorinated polymer (or graphite fluoride, GrF, >61 mass % in F), 2-aminoethanesulfonic acid (Taurine, 99.5%), N,N-dimethylformamide (DMF, 99.8%), acetone (p.a.), ethanol (p.a.), glycerol (99.9 %),

hydrochloric acid (HCl, 0.1 N), sodium hydroxide (NaOH, 0.1 N), were obtained from Sigma-Merck. Deionized water was used for all washings (conductivity ≤ 0.5 μS/cm).

## Catalyst preparation

The catalyst (G-ASA) was prepared by the sulfonation of GrF using taurine in DMF. In a typical procedure, 1 g of GrF (32 mmol in C-F units) was dispersed in 48 mL of DMF in a round-bottom glass flask, stirred for 3 days, and then sonicated (Bandelin Sonorex, DT 255H type, frequency 35 kHz, power 640 W, effective power 160 W) for 4 h. Taurine (4 g, 32 mmol) and $K_2CO_3$ (5.3 g, 1.2 molar excess with respect to taurine) were dissolved separately in 6 mL of ultrapure water. The GrF dispersion in DMF was mixed with a $K_2CO_3$ solution in a round-bottom flask and then added to the taurine solution. $K_2CO_3$ was added to secure basic conditions for keeping taurine's amino group deprotonated and nucleophilic. The mixture was immediately heated to 130 °C under magnetic stirring at 300 rpm for 24 h in an oil bath connected to a reflux condenser. After the mixture cooled, the solid was isolated and washed by centrifugation at 20,000 × g for 8 min. The precipitate was washed several times via centrifugation with solvents (2× hot DMF, 1×DMF, 1×hot acetone, 2× acetone, 3× ethanol, 2× ultrapure water, 2× HCl (2%), and 3× ultrapure water) until the conductivity was below 200 μS/cm. The precipitate was finally redispersed in water and subjected to dialysis for one week until the surrounding water conductivity was below 10 μS/cm. The purified, dispersed product was isolated by centrifugation and acidified by adding 25 wt.% sulfuric acid ($H_2SO_4$ washing was performed to ensure that the taurine molecule's sulfonate group conjugated on the catalyst is protonated, avoiding internal salt formation with the amine group of taurine) and finally washed via centrifugation cycles with methanol followed by freeze drying, and this material was used for characterization and further experiments.

## Catalyst characterization

High-resolution transmission electron microscopy (HR-TEM) and scanning transmission electron microscopy (STEM) in high-angle annular dark-field (HAADF) mode for elemental mapping were performed with an FEI TITAN G2 60-300 HRTEM microscope with an X-FEG type emission gun, operating at 300 kV, objective-lens image spherical aberration corrector, and ChemiSTEM energy-dispersive X-ray spectroscopy (EDS) detector.

X-ray photoelectron spectroscopy (XPS) was carried out with a PHI VersaProbe II (Physical Electronics) spectrometer using an Al Kα source (15 kV, 50 W). The obtained data were evaluated with the MultiPak (Ulvac - PHI, Inc.) software package. The fresh G-ASA catalyst was first treated with the catalytic reaction reagents, thoroughly washed, and then used for the XPS measurements.

Fourier-transform infrared (FT-IR) spectra were recorded on an iS5 FTIR spectrometer (Thermo Nicolet) using the Smart Orbit ZnSe ATR accessory. Briefly, a droplet of ethanol dispersion of the material was placed on a ZnSe crystal and left to dry and form a film. Spectra were acquired by summing 64 scans recorded under a nitrogen gas flow through the ATR accessory. ATR and baseline correction was applied to the collected spectra.

Raman spectra were recorded on a DXR Raman microscope using a diode laser's 613 nm excitation line.

Thermogravimetric analysis (TGA; Netzsch STA 449 C Jupiter thermal analyzer) was performed in synthetic air (100 cm³ min⁻¹). The TGA instrument was equipped with a QMS 403 Aëolos mass spectrometer for evolved gases (EGA). The measurements were carried out using an open crucible made of α-$Al_2O_3$, from 45 °C to 1000 °C, and a heating rate of 10 K min⁻¹. The EGA was focused on m/z 48 and 64 for SO and $SO_2$, respectively. Zeta-potential ($\zeta_p$) measurements were performed with a Zetasizer NanoZS (Malvern UK) instrument on aqueous dispersions of around 0.1 mg mL⁻¹.

Analyses for metal impurities in the samples were performed using inductively coupled plasma mass spectrometry (ICP-MS) with Agilent 7700x (Agilent). A 5 mg sample powder was digested in 1 mL $HNO_3$/HCl mixture (1:3 v/v) with the help of sonication, followed by dilution with water. The dilute solution was analyzed to determine the concentration of metal impurities.

NMR spectra of the esterification products were recorded on a 400 MHz NMR JEOL spectrometer.

The concentration of the acidic sites of the catalyst was determined by acid-base titration. An aqueous NaOH solution (0.05 M, 10 mL) was added to the catalyst (80 mg). Then the mixture was sonicated for 60 min and stirred at room temperature for 12 h. The catalyst was separated by centrifugation, and five milliliters of the supernatant solution were titrated with aqueous HCl solution (0.05 M) using phenol red as an indicator.

## Solketal production

1 g of glycerol and 2.52 g of acetone (1:4 molar ratio) were placed into a 25 mL round-bottom flask and magnetically stirred at room temperature until they formed a homogenous phase. 0.1-0.5 mass% G-ASA catalyst was added to the above mixture, and the stirring continued for 1 h. The product was analyzed by a GC (Agilent 7820 A) equipped with a flame ionization detector (FID). The time-dependent analysis of glycerol conversion related to Supplementary Figure 4 was performed under the same conditions as reported in the main text (glycerol = 1.0 g or 10.85 mmol, acetone = 2.52 g or 43.38 mmol, glycerol: acetone = 1:4, catalyst loading = 0.5 wt% with respect to glycerol). The reactions were carried out individually for different time periods (30, 60 and 120 min) at room temperature. The conversion of glycerol and selectivity of the product was calculated based on the following equations:

$$\text{Conversion of glycerol}\,(\%) = \frac{\text{glycerol converted (mol)}}{\text{initial glycerol (mol)}} \times 100\% \quad (1)$$

$$\text{Selectivity towards solketal}\,(\%) = \frac{\text{solketal formed (mol)}}{\text{glycerol converted (mol)}} \times 100 \quad (2)$$

Turnover frequency (TOF) was calculated according to the following equation:

$$\text{TOF} = \frac{\text{solketal formed (mol)}}{\text{catalyst acidic sites (mol)} \times \text{reaction time (h)}} \quad (3)$$

The number of acidic sites was based on the titration results, which accounted for all the possible active sites, thus providing not-overrated TOF numbers.

Specific productivity was calculated according to the following equation, providing unequivocal reactions rates, free from any inaccuracies in active site estimations, and thus more appropriate for direct comparisons:

$$\text{Specific productivity} = \frac{\text{glycerol converted (mmol)}}{\text{total catalyst amount (g)} \times \text{reaction time (h)}}$$
$$(4)$$

## Recyclability of the catalyst

1 g of glycerol and 2.52 g of acetone were taken into a 25 mL round-bottom flask and magnetically stirred at room temperature until they formed a homogenous phase. 0.5 mass% G-ASA catalyst was quickly added to the above mixture, and the stirring continued for 1 h. The product was analyzed by a GC, and the used catalyst was recovered by centrifugation and washed with acetone several times to remove the impurities adsorbed on the catalyst. The final precipitate was dried at 60 °C overnight before being used for the next cycle.

## Leaching test of the catalyst

Glycerol and acetone were taken at 1:4 mole ratios and transferred into a 25 mL round-bottom flask, and then 0.5 wt% of catalyst was added into the reactant's mixture. The reaction proceeds at room temperature under stirring for 5 min, and the product mixture is isolated from the catalyst. The isolated product mixture was carried out for further reaction, and samples were collected after 1 h and analyzed by gas chromatography.

## Esterification of fatty acid

In a typical experiment, the G-ASA catalyst (10 mg) and fatty acid (0.5 mmol) were mixed in a 2 ml screw-top vial and sonicated for 30 s. Then, under an N₂ atmosphere, methanol (dry, 0.4 ml, ratio methanol/oil 20:1) was added, and the vial was closed with the screw top. The mixture was sonicated for another 30 s, then heated at 60 °C for 4 h. The product was then separated from the catalyst by centrifugation (20000 × g) for 5 min. The supernatant was kept at room temperature for two days and allowed solvent to evaporate for direct NMR analysis.

FG and cyanographene (another FG derivative synthesized as previously reported[55]) were tested as control catalysts to ensure that any contamination from metal traces does not affect the activity. ICP-MS showed that G-ASA and G-CN had similar metal contents (Supplementary Table 4).

## NMR yield calculation

The yield of the esterification reaction was calculated by analyzing the methyl esters in the reaction mixture using quantitative ¹H NMR[64,65]. The methoxy group in the methyl esters at 3.7 ppm (singlet) and the α-carbonyl methylene groups present in the fatty ester derivatives at 2.3 ppm (triplet) are chosen for integration. CDCl₃ solutions of a known amount of fatty acid and methyl esters were used for calibration. The transesterification yield ($Y$) was obtained directly from the area ($A$) of the selected signals:

$$Y\% = 100 \times \frac{2 \times A1}{3 \times A2} \tag{5}$$

where $A1$ and $A2$ are the areas of the methoxy and the methylene protons, respectively.

## Data availability

All data that support the findings of this study are available in the main text, figures, Supplementary Information, and Supplementary data Files. Source data are provided with this paper.

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

## Acknowledgements

A.C.P. acknowledges the support from the European regional development fund (ERDF), European social fund (ESF), and The Ministry of Education, Youth and Sports of the Czech Republic, project no. CZ.02.2.69/0.0/0.0/20_079/0018294. We acknowledge the support by the project Nano4Future (no. CZ.02.1.01/0.0/0.0/16_019/0000754) financed from the ERDF and ESF. A.B. and R.Z. acknowledge the support from the Czech Science Foundation, project no. 19-27454X, EXPRO. M.O. acknowledges the ERC grant 2D-CHEM, No 683024 from H2020. M.M. and M.O. acknowledge the COST Action CA21101. We thank V. Šedajová (XPS), Jana Dzíbelová (TGA-MS), and K. Štymplová (Raman spectroscopy) for the measurements. K.J.R. acknowledges support from Indian Institute of Technology Jammu for providing a seed grant (SGT-100038) and SERB SRG/2020/000865. A.S. thanks IIT Jammu for the PhD Fellowship.

## Author contributions

A.C.P.: Investigation, Analysis, Writing—Original Draft Preparation, Methodology, Visualization; M.M.: Theoretical Investigation, Analysis, Writing; V.R.B.: Investigation; A.S.: Investigation; D.S.: Investigation; S.B.K.: Review; H.B.: Investigation; M.O.: Review & Editing, Funding Acquisition; K.J.: Supervision, Review & Editing; A.B.: Supervision, Writing—Original Draft Preparation, Writing—Review & Editing; R.Z.: Supervision, Funding Acquisition, Review & Editing.

## Competing interests

The authors declare no competing interests.
