## [Peer Review File · Nature Communications]

Acidic Graphene Organocatalysts for Superior Transformation of Wastes into High-Added-Value ChemicalsREVIEWER COMMENTS

Reviewer #1 (Remarks to the Author):

The valorization of glycerol is an important issue. In this manuscript, the authors prepared a special catalyst, graphene catalyst (G-ASA), functionalized with a natural amino acid (taurine), and applied it for the conversion of glycerol to solketal. The catalyst exhibits high activity and high selectivity at room temperature, which is much higher than the previously reported systems. My concerns are the followings:

Introduction part:

1. The discussion about the source of glycerol should be more precise.
2. When discussing the drawbacks of the previously reported systems, the list of the drawbacks should be objective. For example, from Table S1, the lowest selectivity towards solketal reported previously is 92%, while the authors listed "low selectivity" as one short come of the reported systems in the main text (Line 74), however, the lowest selectivity of the present system is 89.3%.
3. For the use of specific productivity to evaluate the performance of a catalyst, the authors are suggested to provide literature citation or more explanation.

Results and discussion

1. When doing deconvolution analysis of the C1s peak, the authors are suggested to consider XPS signal originated from the contamination carbon, usually used to calibrate the others at 284.6 eV, then reconsider the related data, and related analysis and discussion.
2. In Figure 3, please check if the chemical formula of solketal is correct.
3. The authors are suggested to describe if acetal was also quantified, and how about the carbon balance of the catalytic system.
4. When testing the reusability of the catalyst, "the sample was protonated by washing with 25 % sulfuric acid and then washed with methanol to remove excess acid", the authors are suggested to explain why washing by sulfuric acid is needed. If there were data without such pretreatment processes, the reusability of the catalyst would be more persuasive.
5. Concerning the DFT investigation, it is not clear if the authors have carried out IRC calculation to check if the proposed intermediates are really connected by the transition states suggested. The authors are suggested to consider this issue. Otherwise, the related discussion might be miss-leading.

Others: proof reading should be carefully checked. Typical examples are:

1. Line 226, line 256, please check if "Table S2" should be "Table S1".
2. Lines 296-298, please check if the sentence, "In another work, transesterification of tripalmitin to palmitic acid ester was catalyzed by superhydrophobic mesoporous polymers and obtained a yield of 99.9% after a 16-hour reaction at 65 °C.", needs to be improved.

Reviewer #2 (Remarks to the Author):

The authors report their findings on a functionalised graphene catalyst which has been applied to the formation of solketal from glycerol and acetone. The functionalised catalyst has a high activity compared to many other catalysts, helpfully as specific activity as this is not always given in the field, both homogeneous and heterogeneous and as such represents a worthy advance in production of a useful molecule from a bio-sourced waste. Further, the catalyst can be recycled in a simple manner with little to no loss in activity, a point about which is a useful parameter to allow comparison to other recyclable catalysts. Overall, the manuscript is well presented and DFT calculation included along with key characterisation has been reported, both pre- and post-reaction. I would like to know if the taurine is stable over the reaction period and did the authors note any S present in the liquid fraction post-centrifugation? Furthermore, was the reaction monitored over the 1 h time period, is there any significant change in selectivity or reduction in reaction rate over the typical reaction profile seen whereby the rate reduces as the reactants are used up?

The authors apply the catalyst to another process and highlight its efficacy and helpfully compare the ester yield to other similar catalysts. Clearly, this should inspire other groups to consider this type of material. However, two points strike me as being a challenge when discussing new approaches to industrial application. Complicated catalyst preparations and use of crude feed-stocks. The catalyst is an order greater in activity to H₂SO₄ and recyclable without the side-waste described by the authors. This element is not a large hurdle and would be worthwhile to pursue following some analysis of the potential waste vs productivity and economics. The later point is perhaps more pertinent, in that typically glycerol is not available cheaply in a purified form. The biodiesel industry produces a very mixed glycerol waste stream. Tolerance of those additives would be a clear advantage and perhaps worth studying in the future. Could the authors comment on the general robustness of the catalyst, TGA experiments are mentioned and show that the taurine groups are removed >200C which suggests good adhesion? Perhaps this relates to the above points which were not addressed with respect to time-on-line profiles and potential desorption of taurine during reaction. I fully appreciate that the catalyst appears recyclable, however, perhaps the taurine density is such that it can afford to lose an appreciable quantity of active sites and maintain good activity.

In summary, I recommend publication following addressing these minor points.

Minor issue, line 226 Table S2 should be Table S1?

Reviewer #3 (Remarks to the Author):

Poulose et al. performed glycerol acetalization reaction using amino acid functionalized graphene catalyst at ambient conditions. Key points are amino acid functionalization, solvent-free reaction conditions, high specific productivity, and theoretical evidence to explain the catalytic activity. It deserves to publish after major revisions, and following issues should be addressed:

1. In Section 4.2., authors have mentioned "Finally, the dispersion was acidified by 25 wt.% sulfuric acid to secure that all acidic sites are protonated, and finally washed via centrifugation cycles with methanol followed by freeze drying, and this material was used for characterization and further experiments". It is not clear of using H₂SO₄. If the amino acid functionalized graphene catalyst needs to be activated by H₂SO₄, then what is the purpose of using taurine? Sulfonic acid functionalized GO or rGO by H₂SO₄

could be the best choice over the synthesized catalyst. Therefore, authors are requested to compare the catalytic activity with sulfonic acid functionalized GO or rGO.

2. During final dispersion of the catalyst with 25 wt.% sulfuric acid, whether any oxygen and sulfonic acid functionalities incorporate at the edge of the graphene or not need to be confirmed by CHNS/O analysis. In XPS, $-SO_4$ species is also found. How it is attached with the catalyst and has it any role in this reaction?

3. In C (1s) XPS spectrum, the peak corresponds to B.E. approx. 290 eV is for C-F or $\pi-\pi^*$ shake-up satellite peak? As authors stated that almost all F atoms had been eliminated.

4. The authors are asked to make the S (2p) XPS B.E. range uniform for main text and SI.

5. In section 2.2. and Figure 3, authors claimed that they obtained 96.5 % glycerol conversion and 96.8 % solketal selectivity. Reporting activity to one decimal place will be accurate when error bar will be provided.

6. Why lowering the glycerol: acetone mole ratio improves the catalytic activity?

7. During recyclability test, authors protonated the catalyst with H_2SO_4 . However, in the mechanism - SO_3H species remains intact. So, what was the necessity to use H_2SO_4 . The hot filtration test should be performed to check heterogeneity of the catalyst.

8. DFT energies are only accurate when all details will be provided in SI (e.g. frequency, energy, thermal correction, coordinates...).

Point-By-Point Answers to the Reviewers' Comments

Reviewer #1

The valorization of glycerol is an important issue. In this manuscript, the authors prepared a special catalyst, graphene catalyst (G-ASA), functionalized with a natural amino acid (taurine), and applied it for the conversion of glycerol to solketal. The catalyst exhibits high activity and high selectivity at room temperature, which is much higher than the previously reported systems. My concerns are the followings:

Introduction part:

Comment 1.1. The discussion about the source of glycerol should be more precise.

Reply: As per the reviewer's suggestion, we improved the discussion about glycerol with more and clear information, as well as with additional references:

Page 2 of the revised manuscript:

“Large amounts of glycerol are generated by the production of biodiesel^{12,13} as a ten vol.% side-product during the transesterification of triglycerides from animal, vegetable, and algae oils.^{1,14} It can also be produced from biomass fermentation and as a by-product of propylene synthesis or soap production.^{15,16”}

Comment 1.2. When discussing the drawbacks of the previously reported systems, the list of the drawbacks should be objective. For example, from Table S1, the lowest selectivity towards solketal reported previously is 92%, while the authors listed “low selectivity” as one short come of the reported systems in the main text (Line 74), however, the lowest selectivity of the present system is 89.3%.

Reply: It was a factual error from our side to include selectivity as a drawback, and we thank the reviewer for finding this mistake. Indeed, the comparative Table S1 in the Supplementary Information clearly shows that selectivity is not a challenge for this type of reaction. In fact, this is also deduced from the theoretical results, where (page 12 of the original manuscript) it was stated, “*Indeed, as we can see in the comparative Table S1, selectivity is rarely an issue, but mainly the activity of the catalyst.*” fully supporting the comment of the reviewer.

Thus, we removed the selectivity aspect from the list of drawbacks on page 3.

Comment 1.3. For the use of specific productivity to evaluate the performance of a catalyst, the authors are suggested to provide literature citation or more explanation.

Reply: As suggested by the reviewer, we have added the information about the specific productivity (mass-normalized rate of product formation), and its citation (Applied Materials Today 23, 101053, 2021) on page 3. In the experimental section, page 20, it is mentioned that specific productivity is calculated based on the amount (mmol) of product per time unit (h) relative to the mass of the catalyst (g):

$$\text{Specific productivity} = \frac{\text{glycerol converted (mmol)}}{\text{total catalyst amount (g)} \times \text{reaction time (h)}}$$

Comment 1.4. When doing deconvolution analysis of the C1s peak, the authors are suggested to consider XPS signal originated from the contamination carbon, usually used to calibrate the others at 284.6 eV, then reconsider the related data, and related analysis and discussion.

Reply: All XPS data are corrected against the C1s 284.6 eV before deconvolution and plotting. In the present case, since the material (the catalyst itself) is carbon/graphene, we have used the aromatic carbon signal set at 284.6 eV as the reference point for the calibration of all other signals. We have carefully checked all spectra to secure that all calibration and analyses are based on the 284.6 eV peak of aromatic carbon components.

Comment 1.5. In Figure 3, please check if the chemical formula of solketal is correct.

Reply: We thank the reviewer for identifying this mistake in the chemical formula. There was one extra OH in the solketal formula, which should be only O, and is now corrected (Fig 3a, page 10).

Comment 1.6. The authors are suggested to describe if acetal was also quantified, and how about the carbon balance of the catalytic system.

Reply: In the revised manuscript, we have calculated the carbon balance for the catalytic reaction after quantifying all the products, including acetal. The reaction was performed at room temperature with 0.5 wt % of catalyst with respect to glycerol (1.0 g). After 1h, methanol was added to the reaction mixture and the catalyst was separated from reaction mixture using centrifuge. The product mixture was analyzed by gas chromatography. The carbon balance, not considering the excess of acetone, but only the stoichiometric amount i.e. the same as glycerol (10.85 mmol), is 98%. Considering the remaining glycerol and acetal as wastes, and solketal as the only carbon-containing product, then the carbon balance is 93.4%

Comment 1.7. When testing the reusability of the catalyst, “the sample was protonated by washing with 25 % sulfuric acid and then washed with methanol to remove excess acid” the authors are suggested to explain why washing by sulfuric acid is needed. If there were data without such pretreatment processes, the reusability of the catalyst would be more persuasive.

Reply: This is a very good point raised by the reviewer. In the revised manuscript (page 18), we have clarified this and included the following text, explaining the use of sulfuric acid washing.

“H₂SO₄ washing was performed to ensure that the taurine molecule's sulfonate group conjugated on the catalyst is protonated, avoiding internal salt formation with the amine group of taurine.”

The reviewer is correct that since the G-ASA material is a catalyst, after the first treatment for full protonation, there should be no further need to wash the catalyst with H₂SO₄ after each cycle. Therefore, in the revised manuscript, we also performed the catalyst recycling without H₂SO₄ washing steps and received the same performance on both conversion and selectivity.

In the experimental part, page 21 we have modified the washing procedure and highlighted the changes as given below:

“The product was analyzed by GC, and the used catalyst was recovered by centrifugation and washed with acetone several times to remove the impurities adsorbed on the catalyst. The final precipitate was dried at 60 °C overnight before being used for the next cycle.”

Comment 1.8. Concerning the DFT investigation, it is not clear if the authors have carried out IRC calculation to check if the proposed intermediates are really connected by the transition states suggested. The authors are suggested to consider this issue. Otherwise, the related discussion might be miss-leading.

Reply: The reviewer is right that the mechanistic discussion must be based on reliable reaction pathways connecting reactants and products via corresponding transition states (TSs). We paid particular attention to this issue using scans along reaction coordinates, instead of using the IRC approach. Although the IRC approach is a useful tool to investigate reaction pathways, it requires the knowledge of the TS structure including the initial force constants, which then enables to proceed along the direction of the transition vector back and forward towards the reactants and products, respectively. Such approach is appropriate for (rather small) systems, which can be fully relaxed in the geometry optimization steps and for which the evaluation of the Hessian matrix is affordable. It is, however, not suitable for investigating the pathways involving large structures, where the “active region” is attached to a skeleton partially frozen during the optimization step as it is in our case (only the active region was allowed to relax during the optimizations with the carbon lattice kept frozen). In addition, the modeled catalytic reaction takes place in highly acidic environment, which implies the possibility of proton transfer events, in which the environment can be involved. To localize the TSs in such cases is a challenging task even for much smaller systems. Nevertheless, to explore the reaction pathways and estimate the related barriers, we performed a series of back and forward (partially relaxed) scans starting from the reactants and products (or intermediates) in each reaction step along a carefully chosen reaction coordinate. In such an approach, it is assumed that the structure changes its protonated state (i.e., the proton transfer occurs) if the potential energy curve becomes lower along the particular scan. For example, during the formation of an adduct A without the catalyst (Figure S6 in SI), the formation of a C–O bond between glycerol is accompanied by a proton transfer from glycerol to the oxygen atom of acetone, which can however be assisted by the acidic environment. Therefore, the barrier shown in Fig. 6a is estimated from back and forward scans as shown in Fig. 6b. It should also be underlined that the choice of the internal reaction coordinates is in our case always chemically well founded, because it can be presumed that the formation of adducts (steps 1-3 in Figure 4) involves an attack of one of the oxygen atoms of glycerol on the carbonyl group of acetone and also the formation of cyclic products (steps 4-6 in Figure 4) requires approaching specific atoms.

To make this point clear for the reader, we added a short paragraph to Computational details in SI (page 4). The scans based on which the barriers were determined have been added to Figures S7, S9, and S14.

In addition being inspired by the reviewer`s comment, we performed new calculations for systems allowing to localize the TS structure analytically (e.g., for cyclization of protonated adducts A and B), which confirmed that the barriers determined based on relaxed scans were reasonable (cf. Figure S11 and new Table S3 shown below).

Figure S6. Energy diagram (a) and the corresponding relaxed scan along the C(ace)···O(gly) coordinate (b) of the formation of an adduct from acetone and glycerol in acetone with the catalyst. ω B97X-D/6-31+G(d)/SMD(solvent=acetone).

Figure S11. Relaxed scans for the cyclization of protonated adducts A-H⁺ (a) and B-H⁺ (b) along the marked C···O coordinate leading to protonated solketal and acetal, respectively. Computational level: ω B97X-D/6-31+G(d)/SMD(solvent=acetone).

Table S2. The relative electronic and standard Gibbs energies (kcal/mol, $T = 298.15$ K) of the closed form and the TS with respect to the open form of protonated adducts A-H⁺ and B-H⁺ calculated at the ω B97XD/6-31+G(d)/SMD level of theory. Note: The TS structures were fully optimized starting from the maxima obtained by relaxed scans displayed in Figure S11.

	Adduct A-H ⁺		Adduct B-H ⁺	
	ΔE	ΔG°	ΔE	ΔG°
Open form	0.0	0.0	0.0	0.0
Closed form	3.8	7.0	4.0	7.4

Implemented changes:

The following paragraph was added to Computational details in SI page 5:

“It should be emphasized that the modeled catalytic reaction takes place in highly acidic environment, which implies the possibility of proton transfer events, in which the environment can be involved. To estimate the activation barriers, we performed a series of back and forward (partially relaxed) scans starting from the reactants and products (or intermediates) in each reaction step along a carefully chosen reaction coordinate. In such an approach, it was assumed that the structure changed its protonated state (i.e., the proton transfer occurs) if the potential energy curve became lower along the particular scan. For example, during the formation of an adduct A without the catalyst (Figure S6 in SI), the formation of a C=O bond between glycerol was accompanied by a proton transfer from glycerol to the oxygen atom of acetone, which could however be assisted by the acidic environment. Therefore, the barrier shown in Fig. 6a was estimated from back and forward scans as shown in Fig. 6b. It should also be underlined that the choice of the internal reaction coordinates was chemically well founded, because it could be presumed that the formation of adducts (steps 1-3 in Figure 4) involved an attack of one of the oxygen atoms of glycerol on the carbonyl group of acetone and also the formation of cyclic products (steps 4-6 in Figure 4) required approaching specific atoms.”

Figures S7, S9, and S14 were modified to include the scans based on which the barriers were determined:

Figure S7. (a) Energy diagram (in kcal/mol) of the first phase of the catalyzed reaction, i.e. the formation of an adduct A (steps 2a and 3a in Figure 5) along the C-65(acetone)⋯O-86(gly) coordinate (see Figure S8). (b) Relaxed scan along the O⋯H coordinate corresponding to a proton transfer from sulfonic group to an oxygen atom of acetone (step B → C → D in panel a). (c) Relaxed scan along C-65(acetone)⋯O-86(gly) coordinate (step D → E → F in panel a). (d) Relaxed scan along the C-65(acetone)⋯O-86(gly) coordinate (step F → G → H in panel a); the black and red points correspond to structures with the hydrogen atom bonded to the adduct (specifically to an etheric oxygen) and sulfonic group, respectively, indicating that the proton transfer is practically barrierless and occurs in the vicinity of the F minimum. Computational level: ω B97X-D/6-31+G(d)/SMD(solvent=acetone).

Figure S9. (a) Energy diagram (in kcal/mol) of the first phase of the catalyzed reaction, i.e. the formation of an adduct B (steps 2b and 3b in Figure 5) along the C(acetone)⋯O(gly) coordinate (see Figure SX7). (b) Relaxed scan along the O⋯H coordinate corresponding to a proton transfer from sulfonic group to an oxygen atom of acetone (step B → C → D in panel a). (c) Relaxed scan along the C-65(acetone)⋯O-86(gly) coordinate (step D → E → F in panel a). Computational level: ωB97X-D/6-31+G(d)/ SMD(solvent=acetone).

Figure S14. Reaction energy profiles (in kcal/mol) of the alkylsulfonic analog (blue) in comparison to the G-ASA catalyst (black) for the first phase of the reaction: formation of adduct A (steps 2a and 3a in Figure 5) along the C(acetone) \cdots O(gly) coordinate. The structures displayed here represent the alkylsulfonic catalyst analogue. The corresponding structures of G-ASA are shown in Figure S7. (b) Relaxed scan along the O \cdots H coordinate corresponding to a proton transfer from sulfonic group to an oxygen atom of acetone (step B \rightarrow C \rightarrow D in panel a). (c) Relaxed scan along the C-65(acetone) \cdots O-86(gly) coordinate (step D \rightarrow E \rightarrow F in panel a). (d) Relaxed scan along O-63 \cdots H-87 coordinate; proton transfer from an adduct to the sulfonate group (step F \rightarrow G \rightarrow H in panel a). Computational level: ω B97X-D/6-31+G(d)/SMD(solvent=acetone).

A new Table S2 was added to SI:

Table S2. The relative electronic and standard Gibbs energies (kcal/mol, $T = 298.15$ K) of the closed form and the TS with respect to the open form of protonated adducts A-H $^+$ and B-H $^+$ calculated at the

ω B97XD/6-31+G(d)/SMD level of theory. Note: The TS structures were fully optimized starting from the maxima obtained by relaxed scans displayed in Figure S11.

	Adduct A-H ⁺		Adduct B-H ⁺	
	ΔE	ΔG°	ΔE	ΔG°
Open form	0.0	0.0	0.0	0.0
Closed form	3.8	7.0	4.0	7.4
TS	5.2	7.2	4.4	7.2

Comment 1.9. Others: proof reading should be carefully checked. Typical examples are: Line 226, line 256, please check if “Table S2” should be “Table S1”.

Reply: We thank the reviewer highlighting this aspect. We have carefully reviewed the manuscript before submitting the revised version and have corrected the table numbers and other small mistakes.

Comment 1.10. Lines 296-298, please check if the sentence, “In another work, transesterification of tripalmitin to palmitic acid ester was catalyzed by superhydrophobic mesoporous polymers and obtained a yield of 99.9% after a 16-hour reaction at 65 °C.”, needs to be improved.

Reply: We decided to remove the particular example in the revised manuscript because it is about transesterification (page 16). We kept the other examples that refer to esterification, fully matching with the esterification reaction discussed in this manuscript.

Point-By-Point Answers to the Reviewers' Comments

Reviewer #2

The authors report their findings on a functionalised graphene catalyst which has been applied to the formation of solketal from glycerol and acetone. The functionalised catalyst has a high activity compared to many other catalysts, helpfully as specific activity as this is not always given in the field, both homogeneous and heterogeneous and as such represents a worthy advance in production of a useful molecule from a bio-sourced waste. Further, the catalyst can be recycled in a simple manner with little to no loss in activity, a point about which is a useful parameter to allow comparison to other recyclable catalysts. Overall, the manuscript is well presented and DFT calculation included along with key characterisation has been reported, both pre- and post-reaction.

Comment 2.1. I would like to know if the taurine is stable over the reaction period and did the authors note any S present in the liquid fraction post-centrifugation?

Reply: This point is certainly very important. After synthesis of the catalyst, a step involving extensive dialysis removes non-covalently bound taurine molecules from the graphene's surface. Furthermore, the stability of the taurine molecule is evident from the persistence of the sulfur content, according to the XPS spectra, after three (Fig. S3) and five (Fig. S16) recycling reactions, indicating the stability of the catalyst and of the sulfur-containing taurine molecules immobilized on the catalyst. In addition, we have also performed the catalyst leaching test to check and confirm the heterogeneity of the catalyst. According to this, the catalyst was separated and removed from the reaction mixture after 5 min from starting the reaction. The reaction was allowed to continue. However, glycerol and solketal concentrations did not change further, indicating that the G-ASA catalyst is fully heterogeneous, confirming that there is no leaching in the reaction mixture of any catalytically active species from the catalyst's surface. In the revised manuscript, we have described the results of this experiment on page 10:

"To further check the stability and heterogeneity of the catalyst we performed a leaching test, whereby the catalyst was separated from the reaction mixture after 5 min from starting the reaction, after which point no further glycerol conversion was observed by GC, confirming that there is no leaching of any catalytically active species from the catalyst's surface in the reaction mixture."

Please also see our reply to comment 2.4.

Comment 2.2. Furthermore, was the reaction monitored over the 1 h time period, is there any significant change in selectivity or reduction in reaction rate over the typical reaction profile seen whereby the rate reduces as the reactants are used up?

Reply: We have conducted the time resolved study to find the catalyst's activity and glycerol conversion. Glycerol conversion increased from 64.7% (at the first 30 min) to 96.5 % at 60 min, as shown in the figure below. After 120 min of reaction, there is a drop in the glycerol conversion (92.5%) and solketal selectivity (92.1%). Similarly, the reaction rate (specific productivity in this case) is also decreasing (see figure below). The apparent reduction in conversion and is attributed to product hydrolysis by water, which is formed during the reaction (Figure 4 of the manuscript). The

drop in specific productivity (reaction rate), as also the reviewer commented, is attributed to the consumption of the reactants.

Figure description: Time resolved study of the glycerol acetylation reaction. Reaction conditions: glycerol = 1.0 g (10.85 mmol), acetone = 2.52 g, (43.38 mmol) Glycerol: acetone = 1:4, catalyst -0.5 wt% (referred to substrate), reaction carried at R.T for 1h.

Comment 2.3. The authors apply the catalyst to another process and highlight its efficacy and helpfully compare the ester yield to other similar catalysts. Clearly, this should inspire other groups to consider this type of material. However, two points strike me as being a challenge when discussing new approaches to industrial application. Complicated catalyst preparations and use of crude feed-stocks. The catalyst is an order greater in activity to H₂SO₄ and recyclable without the side-waste described by the authors. This element is not a large hurdle and would be worthwhile to pursue following some analysis of the potential waste vs productivity and economics. The later point is perhaps more pertinent, in that typically glycerol is not available cheaply in a purified form. The biodiesel industry produces a very mixed glycerol waste stream. Tolerance of those additives would be a clear advantage and perhaps worth studying in the future.

Reply: We thank the reviewer for the valuable suggestions. Indeed, the tolerance of catalyst to additives, which appear in industrially produced glycerol is an important aspect and must be the focus of future studies.

Comment 2.4. Could the authors comment on the general robustness of the catalyst, TGA experiments are mentioned and show that the taurine groups are removed >200C which suggests good adhesion? Perhaps this relates to the above points which were not addressed with respect to time-on-line profiles and potential desorption of taurine during reaction. I fully appreciate that the catalyst appears recyclable, however, perhaps the taurine density is such that it can afford to lose an

appreciable quantity of active sites and maintain good activity. In summary, I recommend publication following addressing these minor points.

Reply: This point highlighted by the reviewer is very important. To secure that there is no release of taurine units in the reaction medium, we performed the “leaching test”. According to this, the catalyst was separated and removed from the reaction mixture after 5 min from starting the reaction. The reaction was allowed to continue. However, glycerol and solketal concentrations did not change further, indicating that the G-ASA catalyst is fully heterogeneous, confirming that there is no leaching in the reaction mixture of any catalytically active species from the catalyst’s surface. In the revised manuscript, we have described the results of this experiment on page 10:

“To further check the stability and heterogeneity of the catalyst we performed a leaching test, whereby the catalyst was separated from the reaction mixture after 5 min from starting the reaction, after which point no further glycerol conversion was observed by GC, confirming that there is no leaching of any catalytically active species from the catalyst’s surface in the reaction mixture.”

The TGA results indicate as well the strong/covalent bonding of taurine on graphene since the sulfur-containing gasses are released with a maximum rate above 350 °C (Figure 1d in the MS)

Furthermore, on page 8 of the revised manuscript, it is mentioned that:

*“The N 1s core level XPS spectrum (Figure 1f) showed three components at BEs of 399, 400.1, and 401.6 eV, assigned to the secondary non-protonated amine (C-NH-C), to the related hydrogen bonding configurations,⁵⁶ and the protonated⁵⁵ secondary amine groups, respectively. The N 1s XPS core level spectrum of pure taurine (Figure S2) shows a substantial shift for all three N-components at higher eVs in comparison to G-ASA, indicative of the lower electron density of the primary nitrogen in taurine in comparison to the secondary nitrogen in G-ASA, **thus confirming the covalent conjugation of taurine to the graphene support.**”*

Finally, elemental analyses with XPS of the used catalyst after the third cycle for the solketal reaction and after the fifth cycle of the esterification reaction showed that the sulfur content remained unchanged, thus verifying its stability.

In conclusion, the described experiments confirm the stability of the catalyst and the robust immobilization of taurine on the graphene skeleton.

Comment 2.5. Minor issue, line 226 Table S2 should be Table S1?

Reply: We thank the reviewer for identifying this mistake. We have corrected it (Page3).

Point-By-Point Answers to the Reviewers' Comments

Reviewer #3

Poulose et al. performed glycerol acetalization reaction using amino acid functionalized graphene catalyst at ambient conditions. Key points are amino acid functionalization, solvent-free reaction conditions, high specific productivity, and theoretical evidence to explain the catalytic activity. It deserves to publish after major revisions, and following issues should be addressed:

Comment 3.1. In Section 4.2., authors have mentioned “Finally, the dispersion was acidified by 25 wt.% sulfuric acid to secure that all acidic sites are protonated, and finally washed via centrifugation cycles with methanol followed by freeze drying, and this material was used for characterization and further experiments”. It is not clear of using H₂SO₄. If the amino acid functionalized graphene catalyst needs to be activated by H₂SO₄, then what is the purpose of using taurine? Sulfonic acid functionalized GO or rGO by H₂SO₄ could be the best choice over the synthesized catalyst. Therefore, authors are requested to compare the catalytic activity with sulfonic acid functionalized GO or rGO.

Reply: In the revised manuscript we have now clarified this point. H₂SO₄ washing was performed to ensure that the taurine's sulfonate group is protonated, to avoid internal salt formation with the amine group, also contained in taurine's molecule.

The reviewer is correct that since the G-ASA material is a catalyst, after the first treatment for full protonation, there should be no further need to wash the catalyst with H₂SO₄ after each cycle during the recycling and reuse experiments. Therefore, in the revised manuscript, we also performed the catalyst recycling reactions without these washing steps and received the same performance on both conversion and selectivity.

In addition, we have also performed the catalytic reaction with sulfonic acid functionalized rGO (prepared by treating commercially available rGO with 20 % fuming sulfuric acid at 80°C for 24 hours). After the synthesis procedure as described bellow, the rGO-SO₃H catalyst afforded 91% glycerol conversion and 92% solketal selectivity. However, the same catalyst was not active after recycling from the reaction. We thus hypothesized that the rGO-SO₃H was not stable, and active species (e.g. adsorbed sulfonates, or sulfuric acid molecules) leached to the reaction medium. To further verify this, we washed the rGO-SO₃H catalyst three times with hot water (80 °C) and dried it overnight under a vacuum. The hot water-washed rGO-SO₃H catalyst was also inactive for glycerol acetylation. The detailed catalyst preparation procedure is given below.

We also note that such catalysts (sulfonated graphene using H₂SO₄) used in esterification and alcoholysis reactions are active at solvothermal, high pressure conditions at 120 °C (ChemCatChem 6, 2014, 3080-3083), or 80 °C (Fuel, 256, 2019, 115793 115793) or at 90 °C, at ambient pressure in other cases (J. Taiwan Inst. Chem. Engin. 102, 2019, 34-43). In the present case, the reaction is performed at room temperature.

Preparation of rGO-SO₃H

(Adapted: J. Taiwan Inst. Chem. Engin. 102, 2019, 34-43; and from Fuel 256, 2019 115793,

20 ml of oleum 20% (obtained from oleum 65% by diluting with concentrated sulfuric acid) was added to 0.5 g of rGO followed by sonication in an ultrasonic bath for 1 h. The reaction mixture was allowed to stir at 80°C for 24 h under a nitrogen atmosphere. After completion of the reaction, the content was carefully added to double distilled water to convert the unreacted free SO₃ groups to H₂SO₄. This step was performed in an ice-water bath to prevent extreme temperature rise. Then, the resulting mixture was centrifuged, and the obtained solid acid was washed with double distilled water several times to remove the remaining H₂SO₄ and dried overnight under a vacuum.

Chemical source: Sulfuric acid fuming 65% SO₃ (1007201002) and reduced graphene oxide (777684) were purchased from Merck.

Acetylation reaction conditions: glycerol = 10.85 mmol, acetone = 43.38 mmol, catalyst -0.5 wt%, at R.T for 1h

Comment 3.2. During final dispersion of the catalyst with 25 wt.% sulfuric acid, whether any oxygen and sulfonic acid functionalities incorporate at the edge of the graphene or not need to be confirmed by CHNS/O analysis. In XPS, -SO₄ species is also found. How it is attached with the catalyst and has it any role in this reaction?

Reply: The XPS of the G-ASA spectrum after synthesis and purification and before any treatment with sulfuric acid is identical to that after the treatment with sulfuric acid. Therefore, we can be sure that the very mild treatment with sulfuric acid (25 wt% at room temperature) is not contributing to any chemical transformations, but only secures the protonation of the taurine sulfonate group. Like all amino acids, taurine forms internal salts via the protonation of the amino group by the proton from the sulfonate group. The SO₄ species (which are present even before the treatment with the sulfuric acid) originate from slight oxidation of the taurine's sulfonate groups, taking place during the reaction at 120 °C in the presence of the fluorinated graphene.

The -SO₄ species appearing here at 169 eV are also present in other cases where taurine has been immobilized on carbons (<https://www.nature.com/articles/srep21530>; <https://doi.org/10.1016/j.talanta.2019.120356>), but they are not deconvoluted, because only two components are used. The sulfur HR-XPS spectrum goes up to 171 eV in those cases (<https://doi.org/10.1016/j.jtice.2019.05.020>; <https://doi.org/10.1039/C3GC40353J>), as in the present G-ASA case.

Comment 3.3. In C (1s) XPS spectrum, the peak corresponds to B.E. approx. 290 eV is for C-F or π - π^* shake-up satellite peak? As authors stated that almost all F atoms had been eliminated.

Reply: Nearly 1 % fluorine atoms are left in the material, so we assumed that this BE is due to C-F. We agree with the reviewer that π - π^* shake-up satellite peak also appears at this BE. Therefore, in the revised manuscript, we assigned the respective peak both to C-F and π - π^* (Fig. 1e, Page 7).

Comment 3.4. The authors are asked to make the S (2p) XPS B.E. range uniform for main text and SI.

Reply: We thank the reviewer for identifying these discrepancies in S (2p) XPS figures. We have applied these changes and made the axes ranges uniform (Page 3 Fig. 1g).

Comment 3.5. In section 2.2. and Figure 3, authors claimed that they obtained 96.5 % glycerol conversion and 96.8 % solketal selectivity. Reporting activity to one decimal place will be accurate when error bar will be provided.

Reply: We agree with the reviewer. Since not all reactions have at least three trials, we removed the decimal places (Page 10 Fig. 3b&c).

Comment 3.6. Why lowering the glycerol: acetone mole ratio improves the catalytic activity?

Reply: Generally, lowering the glycerol:acetone mole ratio (i.e. increasing acetone), can improve glycerol conversion. However there is probably some point where the excess of acetone leads to volume increase of the reaction mixture diluting the catalyst enough, reducing the reactants' probability of interacting with the active catalyst sites. This effect might be the reason behind the slight drop in conversion, observed in Figure 3b, going from glycerol: acetone ratio of 1:2 (conv. 99.9) to 1:4 (conv. 96.5).

Comment 3.7. During recyclability test, authors protonated the catalyst with H₂SO₄. However, in the mechanism -SO₃H species remains intact. So, what was the necessity to use H₂SO₄. The hot filtration test should be performed to check heterogeneity of the catalyst.

Reply: This point is certainly very important. H₂SO₄ washing was performed to ensure that the taurine molecule's sulfonate group conjugated on the catalyst is protonated and avoids internal salt formation with the amine group that taurine also carries. We have included this point in the manuscript on page 18.

The reviewer is correct that since the G-ASA material is a catalyst, after the first treatment for full protonation, there should be no further need to wash the catalyst with H₂SO₄ after each cycle. Therefore, in the revised manuscript, we also performed the catalyst recycling reactions without these washing steps and received the same performance on both conversion and selectivity.

To check the heterogeneity of the catalyst, we performed a leaching test. According to this, the catalyst was separated and removed from the reaction mixture after 5 min from starting the reaction. The reaction was allowed to continue. However, glycerol and solketal concentrations did not change further, indicating that the G-ASA catalyst is heterogeneous, confirming that there is no leaching in the reaction mixture of any catalytically active species from the catalyst's surface. In the revised manuscript, we have described the results of this experiment on page 10 and the procedure (see below) in the method section, page 20:

“Leaching test of the catalyst: In a typical procedure, glycerol and acetone were mixed at 1:4 mole ratios and transferred into a 25 mL round bottom flask, and then 0.5 wt.% of catalyst was added into the reactant's mixture. The reaction proceeds at room temperature under stirring for 5 minutes, and the product mixture is isolated from the catalyst, and a small amount was analyzed by gas chromatography. The isolated product mixture was left to continue the reaction without the catalyst, and after one hour it was analyzed by gas chromatography.”

Comment 3.8. DFT energies are only accurate when all details will be provided in SI (e.g., frequency, energy, thermal correction, coordinates...).

Reply: We thank the reviewer for this remark. In the revised version, a new file (named xyz coordinates.rar) containing all relevant information (the energies and Cartesian coordinates as well as the thermal corrections and frequencies, if applicable) about each structure presented in the main text and/or the SI file has been included in the submission. We note that the Gibbs energies, thermal corrections and frequencies are only reported for systems, which were fully relaxed during the geometry optimizations, and thus the frequency analysis was meaningful. For the structures which were obtained by constrained optimizations (the active region was only allowed to relax, while the lattice carbon and hydrogen atoms were kept frozen), energies and Cartesian coordinates are provided.

To check the thermal corrections and entropic effects for structures whose size made the frequency analysis affordable, we performed additional calculations. In particular, the conformational analysis of the G-taurine acid models (structures A-G in Fig. S4) shows the relative standard Gibbs energies (at 298.15 K) of various conformers confirming the previous finding that the zwitterion structure is the most stable.

Figure S4. Conformational analysis of G-taurine acid (A-G) and its protonated form (H) in acetone. Relative electronic energies and standard Gibbs energies (reported in parentheses, $T = 298.15$ K) for fully optimized structures were obtained at the ω B97X-D/6-31+G(d)/SMD level of theory. All values are given in kcal/mol.

REVIEWERS' COMMENTS

Reviewer #1 (Remarks to the Author):

The authors have addressed the comments of the reviewers carefully. I recommend acceptance of the manuscript for publication.

Reviewer #2 (Remarks to the Author):

The amendments that the authors have incorporated have improved the manuscript overall. The authors response to comment 2.2 would support the reaction mechanism postulated and would be useful to include in the main text/supplemental information if water is the source of the activity loss. Further, it would be ideal if the reactions were described in terms of if these are individual reactions over 30, 60 and 120 minutes or measurements taken at these time points during one reaction. I recommend acceptance following addressing this minor point.

Reviewer #3 (Remarks to the Author):

The Authors have addressed all points raised by the reviewers' so the revised manuscript is recommended for acceptance

Point-By-Point Answers to the Reviewers' Comments
Reviewer #1

Comment. The authors have addressed the comments of the reviewers carefully. I recommend acceptance of the manuscript for publication.

Reply: We thank the reviewer for appreciating our work and revisions.

Point-By-Point Answers to the Reviewers' Comments
Reviewer #2

Comment. The amendments that the authors have incorporated have improved the manuscript overall. The authors response to comment 2.2 would support the reaction mechanism postulated and would be useful to include in the main text/supplemental information if water is the source of the activity loss. Further, it would be ideal if the reactions were described in terms of if these are individual reactions over 30, 60 and 120 minutes or measurements taken at these time points during one reaction. I recommend acceptance following addressing this minor point.

Reply: We thank the reviewer for appreciating our work and revisions. In the revised manuscript, we have included the requested info by the reviewer, as described below.

Changes applied:

The following text was added on page 9 of the main manuscript and Supplementary Fig. 4. on page 9 of the supplementary information file.

“We executed a time-resolved investigation to evaluate the evolution of product selectivity and reaction rate over time (Supplementary Fig. 4). Results indicate a progression in glycerol conversion from 64.7% at the initial 30 minutes to 96.5% at 60 minutes, followed by a decline to 92.5% and solketal selectivity of 92.1% after 120 minutes of reaction. Similarly, the reaction rate (specific productivity) also exhibited a decline. The observed decrease in conversion is attributed to the hydrolysis of products by water, which is formed during the reaction, while the drop in reaction rate is due to the depletion of reactants.”

In the Experimental Section of the main text, we added on page 17 the following text:

“The time dependent analysis of glycerol conversion related to Supplementary Figure 4 was performed under the same conditions as reported in the main text (glycerol = 1.0 g or 10.85 mmol, acetone = 2.52 g or 43.38 mmol, glycerol: acetone = 1:4, catalyst loading = 0.5 wt% with respect to glycerol). The reactions were carried out individually for different time periods (30, 60 and 120 min) at room temperature.”

Point-By-Point Answers to the Reviewers' Comments
Reviewer #3

Comment. The Authors have addressed all points raised by the reviewers', so the revised manuscript is recommended for acceptance

Reply: We thank the reviewer for appreciating our work and revisions.